# Number of teeth and masticatory function are associated with sarcopenia and diabetes mellitus status among community-dwelling older adults: A Shimane CoHRE study

**Takafumi Abe** [1], **Kazumichi Tominaga**[1,2], **Yuichi Ando**[3], **Yuta Toyama**[1], **Miwako Takeda**[1], **Masayuki Yamasaki**[1,4], **Kenta Okuyama**[1,5], **Tsuyoshi Hamano**[1,6], **Minoru Isomura**[1,4], **Toru Nabika**[1,7], **Shozo Yano** [1,8]*

1 Center for Community-Based Healthcare Research and Education (CoHRE), Organization for Research and Academic Information, Shimane University, Izumo City, Shimane, Japan, 2 Tominaga Dental Office, Ochi-gun, Shimane, Japan, 3 Department of Health Promotion, National Institute of Public Health, Wako City, Saitama, Japan, 4 Faculty of Human Sciences, Shimane University, Matsue City, Shimane, Japan, 5 Center for Primary Health Care Research, Lund University, Malmö, Sweden, 6 Faculty of Sociology, Department of Sports Sociology and Health Sciences, Kyoto Sangyo University, Kita-ku, Kyoto, Japan, 7 Faculty of Medicine, Department of Functional Pathology, Shimane University, Izumo City, Shimane, Japan, 8 Faculty of Medicine, Department of Laboratory Medicine, Shimane University, Izumo City, Shimane, Japan

* syano@med.shimane-u.ac.jp

**Data Availability Statement:** This study protocol, including the consent of the subjects, was approved by the ethics committee of Shimane

## Abstract

### Objectives

We aimed to examine the number of teeth and masticatory function as oral health indices and clarify their roles in the pathogenesis of sarcopenia and diabetes mellitus in community-dwelling older adults.

### Subjects and methods

This cross-sectional study was conducted with 635 older adults in Ohnan, Shimane Prefecture, in rural Japan. The number of teeth and masticatory function (measured by the number of gummy jelly pieces collected after chewing) were evaluated by dental hygienists. Sarcopenia status was assessed using handgrip strength, skeletal muscle index, calf circumference, and a possible sarcopenia diagnosis based on the Asian Working Group for Sarcopenia 2019. Diabetes mellitus status was defined as a hemoglobin A1c level ≥6.5% or self-reported diabetes. A multivariable logistic regression model was used to analyze the association between oral health, sarcopenia, and diabetes mellitus after adjusting for confounders.

### Results

After adjusting for all confounders, logistic regression analysis showed that the number of remaining teeth was negatively associated with a low level of handgrip strength (odds ratio [OR], 0.961; 95% confidence interval [CI], 0.932–0.992) and possible sarcopenia (OR, 0.949; 95% CI, 0.907–0.992). Higher levels of masticatory function were also negatively

University. The consent of the subjects did not include a provision for the data to be shared publicly. Requests for the supporting data can be sent to the ethics committee of Shimane University at kenkyu@med.shimane-u.ac.jp.

**Funding:** This study was supported by the Japan Society for the Promotion of Science (KAKENHI; grant numbers 18K11143, 19K11741 and 19H03996). The funders had no role in study design, data collection and analysis, decision to publish, or preparation of the manuscript.

**Competing interests:** The authors have declared that no competing interests exist.

associated with a low level of handgrip strength (OR, 0.965; 95% CI, 0.941–0.990) and possible sarcopenia (OR, 0.941; 95% CI, 0.904–0.979). Logistic regression analysis showed that the number of remaining teeth and a higher level of masticatory function were negatively associated with diabetes mellitus (OR, 0.978; 95% CI, 0.957–0.999; OR, 0.976; 95% CI, 0.960–0.992, respectively).

## Conclusion

Our findings suggest that improvement in oral health, including the maintenance of masticatory function and remaining teeth, may contribute to the prevention of sarcopenia and diabetes mellitus in older adults.

## Introduction

Oral diseases are extremely prevalent, with more than 3.5 billion individuals affected worldwide [1]. Oral health status is a predictor of cardiovascular disease and all-cause mortality [2,3]. Recently, the concept of oral frailty was proposed based on an integrated oral health status. This includes the number of teeth (NT), chewing ability, articulatory oral motor skill, tongue pressure, and subjective difficulties in eating and swallowing [4]. Oral frailty has been reported to be a risk factor for physical frailty, sarcopenia, disability, and all-cause mortality in a longitudinal study [4]. Although oral health might affect the overall health of an individual, it has been neglected in the public health domain [5].

A recent review reported an association between oral health and sarcopenia [6]. Some studies have reported that the number of remaining teeth and/or objective masticatory performance is related to handgrip strength, skeletal muscle mass, or sarcopenia [4,7–11]. However, because there are few reports and inconsistent results regarding the association between oral health and sarcopenia, further studies are required to address this issue [6].

The potential and bidirectional association between periodontal disease and diabetes mellitus are well known [12–14]. Periodontal disease worsens the oral environment, resulting in a decrease in the NT [15,16], which may affect the diabetic status if masticatory function (MF) is not maintained. A reduction in MF has been reported to be associated with diabetes mellitus [17]. To our knowledge, only one study has examined the association between objective MF and diabetes mellitus.

Recently, numerous bidirectional links between diabetes mellitus and sarcopenia among older adults have been reported. The existence of one condition may increase the risk of developing the other [18]. Therefore, we hypothesized that the maintenance of objective MF and remaining NT as an index of oral health may be related to sarcopenia and diabetes mellitus. The present study aimed to examine the association between oral health status, sarcopenia, and diabetes mellitus among community-dwelling older adults in Japan.

## Materials and methods

### Subjects

This cross-sectional study was part of the Shimane CoHRE study. The Shimane CoHRE study was conducted by Shimane University in collaboration with the annual health examination program that involved the population of Ohnan Town (419 km$^2$, 10,374 people, 44.0% ≥65 years of age, data from the 2015 census) of the Shimane Prefecture in rural Japan between June and July 2017. Annual health examinations are available once a year for residents in this

municipality who are between 40 and 74 years of age and are covered by the National Health Insurance. We provided information regarding this study at least once to potential participants through a document prior to conducting the health examinations. Overall, 852 adults participated in the health examinations. Written informed consent was obtained from 783 participants prior to their enrollment in this study. Seventy-six subjects did not participate in the oral health examination. Subjects with missing data for analyses (n = 72) were excluded; consequently, data from 635 participants were analyzed.

## Ethics

Written informed consent was obtained from all participants. The study protocol was approved by the ethics committee of Shimane University (#2888).

## Data collection

A trained dental hygienist examined the intraoral status of the participants. During the examination, the examiners and participants remained in a seated position, and the number of remaining teeth (excluding third molars and missing teeth) was counted. Objective MF was assessed using a gummy jelly. The participants were instructed to chew the gummy jelly with maximal effort. After 15 s of chewing, the gummy jelly was collected and the number of pieces was counted [19,20].

Handgrip strength was measured in two attempts for each hand. Data were collected based on maximum grip strength. Skeletal muscle mass was measured with a body composition meter using bioimpedance methodology (MC-780A; Tanita Corporation, Tokyo, Japan). The skeletal muscle mass index was estimated based on the trunk and limb muscle mass divided by the square of the body height. Calf circumference was measured twice for both legs. The circumference was the largest in the standing position. Calf circumference was used as the average of the values for both calves. Participants were divided into two groups according to handgrip strength (low or high), skeletal muscle mass index (low or high), and calf circumference (low or high). Cutoff points by sex were used according to the Asian Working Group for Sarcopenia 2019 consensus [21]. Possible sarcopenia (yes or no) was defined based on the assessment protocol [21].

Diabetes mellitus screening was carried out as part of the health examination by measuring serum hemoglobin A1c (HbA1c) levels based on the recommendations of the Japanese Ministry of Health, Labor, and Welfare [22]. Trained nurses or public health nurses assessed the participants according to data obtained from face-to-face structured interviews. This included self-reported physician-diagnosed diabetes mellitus and information regarding the use of hypoglycemic agents. For the present analysis, diabetes mellitus status was defined as an HbA1c value ≥6.5% (NGSP) or self-reported diabetes mellitus [23].

Data on sex (male or female), age, smoking (yes or no; smokers refer to those who have smoked a total of over 100 cigarettes or have smoked over a period of 6 months and have been smoking over the past month), alcohol consumption (no, rarely, sometimes, or daily; assessed from the answer to the question "How often do you drink?"), and physical activity (yes or no; assessed from the answer to the question "In your daily life do you walk or perform any equivalent amount of physical activity for more than 1 hour a day?") were obtained using a questionnaire. Height and weight were objectively assessed as part of the health examination. Body mass index (BMI) was calculated by dividing body weight by height squared (kg/m$^2$).

## Statistics

Frequency data are reported as numbers and percentages, and continuous data are presented as mean ± standard deviation. Multivariable logistic regression analyses were performed to

estimate the odds ratio (OR) and 95% confidence interval (CI) for low levels of handgrip strength, skeletal muscle mass index, calf circumference, or possible sarcopenia with NT (continuous variable) or MF (continuous variable). For all analyses, independent variables were adjusted for sex, age, BMI, smoking, alcohol consumption, and physical activity in Model 1. Model 2 was additionally adjusted for diabetes mellitus. Multivariable logistic regression analyses were performed to estimate the OR and 95% CI for the diabetes mellitus outcome with NT (continuous variable) or MF (continuous variable). For all analyses, independent variables were adjusted for sex, age, BMI, smoking, alcohol consumption, and physical activity in Model 1, and additionally adjusted for possible sarcopenia in Model 2. Statistical analyses were performed using STATA 14.2/IC. All *p*-values for statistical tests were two-tailed, and values <0.05 were regarded as statistically significant.

## Results

### Demographic data of the studied population

Table 1 shows the characteristics of the 635 older adults in this study. In total, 42 (6.6%), 101 (15.9%), 269 (42.4%), and 20 (3.2%) participants had low handgrip strength, low skeletal muscle mass index, low calf circumference, and possible sarcopenia, respectively. The prevalence of diabetes was 17.6%.

### Association between the loss of mastication and sarcopenia

Table 2 shows the association between NT and MF with respect to oral health and systemic sarcopenia status. In Model 2 (all adjusted model), NT was associated with low handgrip strength (OR = 0.961; 95% CI, 0.932–0.992) and possible sarcopenia (OR = 0.949; 95% CI, 0.907–0.992). However, no associations were found between NT and skeletal muscle mass index or calf circumference. In addition, MF was associated with low handgrip strength (OR = 0.965; 95% CI, 0.941–0.990) and possible sarcopenia (OR = 0.941; 95% CI, 0.904–0.979) in Model 2. No associations were found between MF and skeletal muscle mass index or calf circumference.

### Association of the remaining teeth and mastication with diabetes

Table 3 shows the association between NT or MF with respect to oral health status and diabetes mellitus status. In Model 2 (all adjusted model), NT was associated with diabetes mellitus status (OR = 0.978; 95% CI, 0.957–0.999). MF was also associated with diabetes mellitus status (OR = 0.976; 95% CI, 0.976–0.992) in Model 2.

## Discussion

This study suggests that reduced NT or MF is associated with both sarcopenia and diabetes. We observed that a low level of MF was significantly associated with a decline in handgrip strength, possible sarcopenia, and higher odds of diabetes after adjusting for all confounders. The absence of remaining teeth was also associated with a decline in handgrip strength, possible sarcopenia, and higher odds of diabetes after adjusting for all confounders. Thus, our findings suggest that improvement in oral health, including the maintenance of MF and remaining teeth, may contribute to the prevention of sarcopenia and diabetes mellitus in older adults [24].

Our findings are consistent with the results of previous studies, suggesting that a lower MF is associated with reduced handgrip strength as an indicator of sarcopenia. In the present study, participants with a higher MF tended to have lower odds of declining skeletal muscle mass index and calf circumference. However, skeletal muscle mass index and calf

**Table 1. Participants' characteristics.**

| Variables | Total, N = 635 | |
|---|---|---|
| | **n** | **% or SD** |
| Sex | | |
| Male, n (%) | 280 | 44.1 |
| Female, n (%) | 355 | 55.9 |
| Age, mean ± SD | 67.3 | 7.7 |
| Body mass index (kg/m$^2$), mean ± SD | 22.8 | 3.2 |
| Smoking | | |
| No, n (%) | 575 | 90.6 |
| Yes, n (%) | 60 | 9.4 |
| Alcohol consumption | | |
| No, n (%) | 325 | 51.2 |
| Sometimes, n (%) | 129 | 20.3 |
| Daily, n (%) | 181 | 28.5 |
| Physical activity | | |
| Yes, n (%) | 309 | 48.7 |
| No, n (%) | 326 | 51.3 |
| **Oral health status** | | |
| Number of teeth[a], mean ± SD | 21.8 | 9.4 |
| Masticatory function[b], mean ± SD | 26.0 | 13.6 |
| **Sarcopenia status[a]** | | |
| Handgrip strength (kg), mean ± SD | 30.2 | 8.2 |
| Men: <28 kg, Women: <18 kg, n (%) | 42 | 6.6 |
| Skeletal muscle mass (kg/m$^2$), mean ± SD | 7.1 | 1.2 |
| Men: <7.0 kg/m$^2$, Women: <5.7 kg/m$^2$, n (%) | 101 | 15.9 |
| Calf circumference (cm), mean ± SD | 34.0 | 3.0 |
| Men: <34 cm, Woman: <33 cm, n (%) | 269 | 42.4 |
| Possible sarcopenia: yes, n (%) | 20 | 3.2 |
| **Diabetes mellitus status** | | |
| HbA1c, ≥6.5% or self-reported diabetes mellitus, n (%) | 112 | 17.6 |
| HbA1c (%), mean ± SD | 6.0 | 0.6 |
| Self-reported diabetes mellitus | | |
| No, n (%) | 558 | 87.9 |
| Yes, n (%) | 77 | 12.1 |

[a]Sarcopenia status was categorized based on cutoff values per sex according to the study by Chen et al. [21].

HbA1c, hemoglobin A1c; SD, standard deviation.

circumference were not significantly associated with MF. Previous studies have shown that lower masticatory performance is associated with sarcopenia in older adults [4,11]. To the best of our knowledge, only one study has reported that handgrip strength is positively associated with masticatory performance measured with gummy jellies [7]. A recent review speculated that the mechanisms of the association between oral health and sarcopenia involve three pathways in which worsening of oral health causes poor dietary intake, neuromuscular system failure, and a loss of muscle strength caused by inflammation [6]. Meanwhile, we found associations between NT and a decline in handgrip strength, or possible sarcopenia, after adjusting for all confounders. However, higher levels of NT were shown to have lower odds of declining skeletal muscle mass index and calf circumference. A reduction in NT in older adults is treated

**Table 2.  Association between oral health status and sarcopenia status among community-dwelling Japanese adults (N = 635).**

| Oral health status | Handgrip strength[a] | Skeletal muscle mass[a] | Calf circumference[a] | Possible sarcopenia[b] |
|---|---|---|---|---|
| | OR (95% CI) | OR (95% CI) | OR (95% CI) | OR (95% CI) |
| Number of teeth | | | | |
| Model 1 | **0.958 (0.929–0.988)** | 0.979 (0.951–1.008) | 0.980 (0.957–1.002) | **0.951 (0.910–0.994)** |
| Model 2 | **0.961 (0.932–0.992)** | 0.980 (0.951–1.010) | 0.980 (0.957–1.004) | **0.949 (0.907–0.992)** |
| Masticatory function | | | | |
| Model 1 | **0.963 (0.939–0.988)** | 0.981 (0.959–1.003) | 0.985 (0.969–1.000) | **0.942 (0.906–0.980)** |
| Model 2 | **0.965 (0.941–0.990)** | 0.982 (0.960–1.004) | 0.986 (0.970–1.001) | **0.941 (0.904–0.979)** |

Each sarcopenia outcome was analyzed with oral health using the logistic regression after adjusting for sex, age, body mass index, smoking, alcohol consumption, and physical activity in Model 1, or Model 1 plus diabetes mellitus status in Model 2. Values in boldface indicate significance ($p < 0.05$).

OR, odds ratio; CI, confidence interval.

[a]Handgrip strength (men: <28 kg, women: <18 kg), skeletal muscle mass (men: <7.0 kg/m$^2$, women: <5.7 kg/m$^2$), and calf circumference (men: <34 cm, woman: <33 cm) were categorized based on cutoff values according to the study by Chen et al. [21].

[b]Possible sarcopenia was defined based on the assessment protocol according to the study by Chen et al. [21].

by providing dentures with the intervention of a dentist. The use of dentures may mitigate the risk of losing the remaining teeth. This would result in an improvement in MF. Therefore, in this study, MF was measured as an integrated index of elements, including the remaining teeth and the use of dentures [24]. We did not compare MF with or without dentures. Therefore, it is important to examine the associations of MF with dentures in future studies [25].

MF and NT were significantly associated with diabetes mellitus, defined as HbA1c ≥6.5% or self-reported diabetes mellitus, respectively. Yamazaki et al. reported an association between diabetes and MF, which was evaluated using color-changing chewing gum [17]. Their study found that the highest masticatory performance was associated with lower odds of diabetes in Japanese men (OR = 0.53; 95% CI, 0.31–0.90). Our findings support the negative association between MF and diabetes, as reported by Yamazaki et al. [17]. A previous longitudinal study reported that the number of missing teeth (≥15 teeth) was positively associated with the occurrence of new-onset diabetes (hazard ratio = 1.21; 95% CI, 1.09–1.33) among 188,013 Korean adults [26]. This was consistent with our findings that individuals with an increasing number of remaining teeth tended to have lower odds of developing diabetes. We speculated that the

**Table 3.  Association between oral health status and diabetes mellitus status among community-dwelling Japanese adults (N = 635).**

| Oral health status | HbA1c, ≥6.5% or self-reported DM |
|---|---|
| | OR (95% CI) |
| Number of teeth | |
| Model 1 | **0.978 (0.957–0.999)** |
| Model 2 | **0.978 (0.957–0.999)** |
| Masticatory function | |
| Model 1 | **0.976 (0.960–0.992)** |
| Model 2 | **0.976 (0.960–0.992)** |

Diabetes mellitus outcome was analyzed with oral health using the logistic regression after adjusting for sex, age, body mass index, smoking, alcohol consumption, and physical activity in Model 1, or Model 1 plus possible sarcopenia in Model 2. Values in boldface indicate significance ($p < 0.05$).

CI, confidence interval; DM, diabetes mellitus; HbA1c, hemoglobin A1c; OR, odds ratio.

association between oral health and diabetes could be explained as follows. In cases of reduced MF or NT, the increase in soft sugar-rich meals and the shortening of masticatory (meal) time increases postprandial blood glucose levels by insufficient secretion of insulin [27–30]. Deterioration of the oral environment (the morbidity of periodontal disease) leads to decreased insulin sensitivity and impaired glucose tolerance. This results in the development of diabetes [31]. We demonstrated a significant association between MF and sarcopenia as well as diabetes mellitus. Loss of muscle mass, which is the tissue that stores glucose after absorption from the intestine, reduces glucose uptake into skeletal muscle, resulting in an increase in postprandial blood glucose [32,33]. Unhealthy habits such as smoking, alcohol consumption, and lack of physical activity as covariates might have affected the diabetic status independently, as well as the lack of teeth. Participants without teeth may have used dentures. Denture use might have improved MF and nutritional status, and may have affected their dietary selections [34,35]. Future studies are required to assess the use of dentures.

This study has several limitations. First, we used a cross-sectional design, which precludes the possibility of causal inference among oral health, sarcopenia, or diabetes. Due to the bidirectional relationship between diabetes mellitus and sarcopenia among older adults, the existence of one condition may increase the risk of developing the other [18]. Therefore, our study could not explain the causal relationships among oral health, sarcopenia, and diabetes mellitus. Second, the small sample size may have yielded a low statistical power. Third, our study could not control for the effects of the unmeasured factors. These include the effects of oral factors (e.g., periodontal disease, denture use, and brushing teeth) on the relationship between oral health and sarcopenia or diabetes [14,26,36]. Thus, future longitudinal studies are essential for investigating these associations.

In conclusion, among community-dwelling older adults in rural Japan, an improvement in oral health, including the maintenance of MF and remaining NT, may contribute to the prevention of systemic health status deterioration. This decline in health status is associated with sarcopenia and diabetes.

## Acknowledgments

The authors appreciate all members of the public health division of Ohnan Town who assisted with the community health examinations and the CoHRE study members for their skillful assistance.

## Author Contributions

**Conceptualization:** Kazumichi Tominaga.

**Data curation:** Takafumi Abe, Kazumichi Tominaga, Miwako Takeda, Kenta Okuyama.

**Formal analysis:** Takafumi Abe, Kazumichi Tominaga, Kenta Okuyama.

**Funding acquisition:** Takafumi Abe.

**Investigation:** Takafumi Abe, Miwako Takeda, Masayuki Yamasaki.

**Methodology:** Takafumi Abe, Tsuyoshi Hamano.

**Resources:** Minoru Isomura.

**Supervision:** Minoru Isomura, Toru Nabika, Shozo Yano.

**Writing – original draft:** Takafumi Abe, Yuta Toyama.

**Writing – review & editing:** Yuichi Ando, Masayuki Yamasaki, Kenta Okuyama, Tsuyoshi Hamano.

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
