## [Decision Letter · Decision Letter 0]

31 Mar 2021

PONE-D-21-01645

Reduced masticatory function is associated with sarcopenia and diabetes mellitus status among community-dwelling elderlies: Shimane CoHRE study

PLOS ONE

Dear Dr. Yano,

Thank you for submitting your manuscript to PLOS ONE. After careful consideration, we feel that it has merit but does not fully meet PLOS ONE’s publication criteria as it currently stands. Therefore, we invite you to submit a revised version of the manuscript that addresses the points raised during the review process.

Reviewer 1 comes up with some very unerring points. This includes that the paper shows an association between diabetes, and masticatory function, but it needs to be very clearly discussed and pointed out that this does not allow to assume any cause-consequence relationship. In this context, the reviewers´ request to analyse, whether mastictory function is associated with sarcopenia and diabetes *independently*, apperas crucial. With regard to group size, the authors may want to condider an anlysis based on segmentation into less, but larger groups.

We look forward to receiving your revised manuscript.

Kind regards,

Clemens Fürnsinn, Ph.D.

Academic Editor

PLOS ONE

Journal Requirements:

Reviewers' comments:

Reviewer's Responses to Questions

**Comments to the Author**

1. Is the manuscript technically sound, and do the data support the conclusions?

Reviewer #1: No

Reviewer #2: Yes

2. Has the statistical analysis been performed appropriately and rigorously? 

Reviewer #1: No

Reviewer #2: Yes

3. Have the authors made all data underlying the findings in their manuscript fully available?

Reviewer #1: Yes

Reviewer #2: No

4. Is the manuscript presented in an intelligible fashion and written in standard English?

Reviewer #1: No

Reviewer #2: No

5. Review Comments to the Author

Reviewer #1: The study investigated the association between number of teeth/masticatory function and sarcopenia/diabetes. The issue is interesting enough, however, the several important problems should be raised.

1. Sarcopenia and DM have bidirectional association. However, the authors did not mention it in introduction and discussion.

2. Because of mutual association between sarcopenia and DM, these variables should be considered as so-founding factors. It should be investigated if number of teeth/masticatory function is associated with sarcopenia independently with DM or not.

3. The number of sarcopenic subjects was too small to be analyzed. The authors segmented 4 or 5 groups according to number of teeth/masticatory function. For example the number of subjects with low handgrip and NT-G3 or MF-Q3 was only 5. I do not think it is too small for this type of statistical analysis.

Reviewer #2: This is an interesting and significant contribution. The study is well designed and limitations are acknowledged. I recommend having the manuscript revised for language improvement to make it more accurate and engaging. I also recommend you to replace the term elderlies with older adults.

6. PLOS authors have the option to publish the peer review history of their article (what does this mean?). If published, this will include your full peer review and any attached files.

Reviewer #1: No

Reviewer #2: No

---

## [Author Response · Author response to Decision Letter 0]

3 May 2021

Dr. Clemens Fürnsinn

Dear Editor: 

We wish to re-submit the manuscript titled “Number of teeth and masticatory function are associated with sarcopenia and diabetes mellitus status among community-dwelling older adults: a Shimane CoHRE study.” We thank you and the reviewers for your thoughtful suggestions and insights. The manuscript has benefited from these insightful suggestions. In addition, we will change the online submission form on your behalf.

The manuscript has been rechecked and the necessary changes have been made in accordance with the reviewers’ suggestions. The responses to all comments have been prepared and attached herewith. In accordance with comments from Reviewer 1, we have changed the statistical analysis methods. The number of teeth/masticatory functions was used as the continuous variable as the number of cases in each categorized group was small. In addition, in accordance with suggestions from Reviewer 2, we have replaced the term “elderly” with the term “older adults” throughout the manuscript including the title. All the revisions made as per your suggestions have been indicated with track changes in the manuscript. 

Thank you for your consideration. I look forward to hearing from you.

Sincerely,

Shozo Yano 

Department of Laboratory Medicine, Faculty of Medicine

Shimane University

Izumo City, Shimane

Japan.

---

## [Decision Letter · Decision Letter 1]

11 May 2021

PONE-D-21-01645R1

Number of teeth and masticatory function are associated with sarcopenia and diabetes mellitus status among community-dwelling older adults: a Shimane CoHRE study

PLOS ONE

Dear Dr. Yano,

Thank you for submitting your manuscript to PLOS ONE. After careful consideration, we feel that it has merit but does not fully meet PLOS ONE’s publication criteria as it currently stands. Therefore, we invite you to submit a revised version of the manuscript that addresses the points raised during the review process.

There seems to be one relevant point that still needs to be clarified: This is that reviewer 1 asks, why you have used different statistical procedures for different parameters. Please briefly explain this to the readers in the Methods section. If appropriately done, the paper should be acceptable.

We look forward to receiving your revised manuscript.

Kind regards,

Clemens Fürnsinn, Ph.D.

Academic Editor

PLOS ONE

Journal Requirements:

Reviewers' comments:

Reviewer's Responses to Questions

**Comments to the Author**

1. If the authors have adequately addressed your comments raised in a previous round of review and you feel that this manuscript is now acceptable for publication, you may indicate that here to bypass the “Comments to the Author” section, enter your conflict of interest statement in the “Confidential to Editor” section, and submit your "Accept" recommendation.

Reviewer #1: (No Response)

2. Is the manuscript technically sound, and do the data support the conclusions?

Reviewer #1: No

3. Has the statistical analysis been performed appropriately and rigorously? 

Reviewer #1: No

4. Have the authors made all data underlying the findings in their manuscript fully available?

Reviewer #1: No

5. Is the manuscript presented in an intelligible fashion and written in standard English?

Reviewer #1: Yes

6. Review Comments to the Author

Reviewer #1: I do not understand why the authors used Poisson regression analysis. The reasons why they used this method for handgrip and sarcopenia and logistic regression for SMI and calf cir are unclear. These should be clearly explained.

7. PLOS authors have the option to publish the peer review history of their article (what does this mean?). If published, this will include your full peer review and any attached files.

Reviewer #1: No

---

## [Author Response · Author response to Decision Letter 1]

17 May 2021

Response to Reviewer 1

Reviewer #1

Comment 1

I do not understand why the authors used Poisson regression analysis. The reasons why they used this method for handgrip and sarcopenia and logistic regression for SMI and calf cir are unclear. These should be clearly explained.

Reply:

Thank you for your important suggestions. We noticed a mistake in selecting the statistical model based on your comments. A previous study showed that correction of the odds ratio may be desirable to interpret the magnitude of an association when the incidence of outcome is more than 10% and the odds ratio is more than 2.5 or less than 0.5 (Zhang J, Yu KF. JAMA. 1998. PMID: 9832001). In this case, another study suggested using Poisson regression with robust variance (Barros AJ, et al. BMC Med Res Methodol. 2003. PMID: 14567763). However, our study did not meet this criterion. Therefore, we reanalyzed the data using logistic regression and revised the methods, results including Table 2, and the abstract. The results and conclusions did not change after the correction.

<Revised> Page 7, lines 120–124 (Methods) 

Multivariable logistic regression analyses were performed to estimate the odds ratio (OR) and 95% confidence interval (CI) for low levels of handgrip strength, skeletal muscle mass index, calf circumference, or possible sarcopenia with NT (continuous variable) or MF (continuous variable). For all analyses, independent variables were adjusted for sex, age, BMI, smoking, alcohol consumption, and physical activity in Model 1. Model 2 was additionally adjusted for diabetes mellitus.

<Revised> Page 9, lines 140–146 (Results) 

Table 2 shows the association between NT and MF with respect to oral health and systemic sarcopenia status. In Model 2 (all adjusted model), NT was associated with low handgrip strength (OR = 0.961; 95% CI, 0.932–0.992) and possible sarcopenia (OR = 0.949; 95% CI, 0.907–0.992). However, no associations were found between NT and skeletal muscle mass index or calf circumference. In addition, MF was associated with low handgrip strength (OR = 0.965; 95% CI, 0.941–0.990) and possible sarcopenia (OR = 0.941; 95% CI, 0.904–0.979) in Model 2. No associations were found between MF and skeletal muscle mass index or calf circumference.

---

## [Editor Report · Decision Letter 2]

19 May 2021

Number of teeth and masticatory function are associated with sarcopenia and diabetes mellitus status among community-dwelling older adults: a Shimane CoHRE study

PONE-D-21-01645R2

Dear Dr. Yano,

We’re pleased to inform you that your manuscript has been judged scientifically suitable for publication and will be formally accepted for publication once it meets all outstanding technical requirements.

Kind regards,

Clemens Fürnsinn, Ph.D.

Academic Editor

PLOS ONE
---

## [Editor Report · Acceptance letter]

24 May 2021

PONE-D-21-01645R2 

Number of teeth and masticatory function are associated with sarcopenia and diabetes mellitus status among community-dwelling older adults: a Shimane CoHRE study 

Dear Dr. Yano:

I'm pleased to inform you that your manuscript has been deemed suitable for publication in PLOS ONE. Congratulations! Your manuscript is now with our production department. 

Kind regards, 

on behalf of

Prof. Dr. Clemens Fürnsinn 

Academic Editor

PLOS ONE